# Comparative Analysis of Genotyping by Sequencing and Whole-Genome Sequencing Methods in Diversity Studies of *Olea europaea* L.

**DOI:** 10.3390/plants10112514

**Published:** 2021-11-19

**Authors:** James Friel, Aureliano Bombarely, Carmen Dorca Fornell, Francisco Luque, Ana Maria Fernández-Ocaña

**Affiliations:** 1Dipartimento di Bioscienze, Università degli Studi di Milano, 20122 Milan, Italy; james.friel@unimi.it (J.F.); abombarely@ibmcp.upv.es (A.B.); 2Instituto de Biologıa Molecular y Celular de Plantas (IBMCP), CSIC, Universitat Politecnica de Valencia, 46011 Valencia, Spain; 3Departamento de Didáctica de las Matemáticas y las Ciencias Experimentales, Facultad de Educación, Universidad Internacional de la Rioja (UNIR), 26006 Logroño, Spain; mariadelcarmen.dorcafornell@unir.net; 4Instituto Universitario de Investigación en Olivar y Aceites de Oliva (INUO), Universidad de Jaén, 23071 Jaén, Spain; fjluque@ujaen.es; 5Departamento de Biología Animal, Biologia Vegetal y Ecología, Facultad de Ciencias Experimentales, Campus de Las Lagunillas s/n, Universidad de Jaén UJA, 23071 Jaén, Spain

**Keywords:** *Olea europaea* L., olive, genotype by sequencing (GBS), single-nucleotide polymorphism (SNP), whole-genome sequencing (WGS), reference genome

## Abstract

Olive, *Olea europaea* L., is a tree of great economic and cultural importance in the Mediterranean basin. Thousands of cultivars have been described, of which around 1200 are conserved in the different olive germplasm banks. The genetic characterisation of these cultivars can be performed in different ways. Whole-genome sequencing (WGS) provides more information than the reduced representation methods such as genotype by sequencing (GBS), but at a much higher cost. This may change as the cost of sequencing continues to drop, but, currently, genotyping hundreds of cultivars using WGS is not a realistic goal for most research groups. Our aim is to systematically compare both methodologies applied to olive genotyping and summarise any possible recommendations for the geneticists and molecular breeders of the olive scientific community. In this work, we used a selection of 24 cultivars from an olive core collection from the World Olive Germplasm Collection of the Andalusian Institute of Agricultural and Fisheries Research and Training (WOGBC), which represent the most of the cultivars present in cultivated fields over the world. Our results show that both methodologies deliver similar results in the context of phylogenetic analysis and popular population genetic analysis methods such as clustering. Furthermore, WGS and GBS datasets from different experiments can be merged in a single dataset to perform these analytical methodologies with proper filtering. We also tested the influence of the different olive reference genomes in this type of analysis, finding that they have almost no effect when estimating genetic relationships. This work represents the first comparative study between both sequencing techniques in olive. Our results demonstrate that the use of GBS is a perfectly viable option for replacing WGS and reducing research costs when the goal of the experiment is to characterise the genetic relationship between different accessions. Besides this, we show that it is possible to combine variants from GBS and WGS datasets, allowing the reuse of publicly available data.

## 1. Introduction

Olive tree (*Olea europaea,* ssp. *europaea*, var. *europaea*) is a member of the *Oleaceae* family which has an estimated 600 species of mostly small trees and shrubs [1,2]. Within the genus *Olea*, there are around 35 species and subspecies classified in three subgenera, *Olea, Paniculatae*, and *Tetrapilus* [1,3], *Olea europaea* being the most widely cultivated and economically important species [1,4]. It is a long-lived, outcrossing species of fruit tree native to the Mediterranean basin. Nevertheless, its popularity as a commodity has extended its cultivation to other areas such as the coast of California (United States), the central coast of Chile, southern Africa, southwestern Australia, and Asia [1]. All subspecies are diploid (2n = 2x = 46), with the exception of two inter-subspecies polyploids [5]. As one of the world’s oldest crops, the long history of cultivation and trade has made *O. europaea* culturally and economically significant to many countries in the Mediterranean basin [6]. The wild ancestor was domesticated around 6000 BC in the eastern region of the Mediterranean, but it has since spread across the world [6,7,8,9]. Though the exact origins of its domestication and distribution are unclear, it was likely spread east to west by humans through migration and the trade routes of the Levant area of the Mediterranean, contributing to its genetic diversity [10,11,12].

Over the past 50 years, agricultural intensification and global economics have contributed to a shift towards reliance on a small subset of high-performance cultivars, placing greater reliance on a limited supply of genetic resources [13,14,15]. Therefore, ex situ curation is becoming increasingly necessary for the maintenance and understanding of the genetic resources available for future breeding programs of insertion in the fields of new cultivars and their cultural conservation. To this end, in 1994 the International Olive Council (IOC) began to establish ex situ collections in national and international germplasm banks. Currently, there are three international germplasm banks located in Córdoba, Marrakech, and Izmir, as well as 19 national collections [14,16]. Since their establishment, limited molecular analysis has been carried out on olive germplasm banks to understand the genetic diversity, establish core collections, or suggest progenitors for future breeding programs.

There are around 1200 native cultivars, 3000 synonyms [8], and currently there are 5 published *Olea europaea* assemblies comprising 3 different cultivars, *Olea europaea* cv. ‘Farga’ (Oe6/Oe9) [17,18], *Olea europaea* cv. ‘Picual’ (Oleur0.6.1) [13] and *Olea europaea* cv. ‘Arbequina’ (Oe_Rao) [19], and a purported wild variety *Olea europaea ssp. sylvestris* (Oe451) [20]. Due to the prohibitive cost and effort involved in whole-genome sequencing, it was once an endeavour only available to consortia and solely focused on model organisms. However, the rapidly evolving sequencing technologies from companies such as Illumina, Pacific Biosystems, and Oxford Nanopore have consistently surpassed Moors law [21], allowing even small labs to afford to sequence their own species of interest. This affordable access to rapid high-throughput sequencing has been revolutionary for many areas of biology, with applications in de novo genome assembly, genotyping, gene–trait associations, metagenomics, transcriptomics, and epigenetics [22,23]. Given the economic and cultural importance of olive, it is not surprising that we have seen so many olive genome-sequencing projects in a short space of time. Each of the available olive genome assemblies uses different sequencing technologies, or a combination of methods. It is necessary when choosing a sequencing approach to consider the application of the genome, as the different types of errors, error rates, and biases that can come from a particular methodology will affect the overall quality and completeness of the assembly [24]. Furthermore, the different choices of assembly tool and pipeline will impact on the contiguity, accuracy, and handling of repeat regions in highly polyploid species [25].

Genetic profiling in many crops, as well as the analysis of genetic variation within and between their populations, has been achieved using cheap and effective biochemical and DNA markers such as random amplified fragment polymorphic DNA (RAPDs) [26,27], amplified fragment length polymorphisms (AFLPs) [28,29], SSRs [10,15,30,31], and single-nucleotide polymorphisms (SNPs) [32,33,34]. A sequence assembly of any quality is not required for many projects such as genetic profiling, establishing genetic relatedness, QTL mapping, or to perform a GWAS, although access to a complete and correctly annotated genome assembly provides the best account of individual genome variation and provides more information, increasing the potential resolution when using methods capable of recovering a higher density of variants, such as genotype by sequencing (GBS) [35], or whole-genome sequencing (WGS).

Both GBS and WGS could be used to call SNPs but differ massively in terms of missing data and their cost effectiveness. The main difference is that GBS is a reduced-representation approach to sequencing, that while quick and cost-effective, results in much more missing data due to its DNA fragmentation step. GBS uses restriction-site-specific digestion enzymes to fragment DNA samples. The DNA fragments then have unique barcode sequences ligated to the ends of the DNA fragments before fragment size selection is performed. The primary advantage of GBS is that by assigning sample-specific barcoded adapters, it is possible perform multiplexed sequencing in a single Illumina flow-cell lane for a large number of samples [35], making it much more cost effective than WGS. The number of SNPs that can be identified from within a WGS dataset can be significantly more than with GBS; however, this level of resolution is not always necessary in genetic-linkage-based research [36]. GBS has already been used in olives to generate genetic maps [20,37,38], study the diversity, and perform association analysis [39]. While the GBS library preparation itself is relatively simple, demultiplexing of the raw data is required to process the samples. This step can add extra difficulty for any researcher not already familiar with data processing.

It is likely that as the cost of sequencing and data processing continues to fall, we will see even more cultivar genomes assembled using differing technologies and assembly pipelines. It then becomes important to be able to assess the quality and functionality of a genome in order to choose the right assembly for a project’s goal. Several tools and methods already exist to estimate genome completeness and contiguity [40]; however, the potential impact and bias that may arise from a reference genome’s genetic background is not fully understood, nor to what extent this issue may affect different types of analysis.

In the context of a typical population genetics study, we wanted to understand if the genetic background would have a significant impact on the interpretation of group clustering and genetic relationships. Given the difference in data production and cost, we asked how comparable GBS datasets are to WGS datasets for this mode of research. Furthermore, we assessed the practicality of combining WGS and GBS datasets (Table 1), as this would allow for unrelated sequencing projects to access and combine publicly available data using two very different genotyping methods.

## 2. Results and Discussion

### 2.1. Genome Assembly Comparison

In this study, we used five publicly available *Olea europaea* subsp. *europaea* assemblies (Table 2) comprising three different cultivars, ‘Farga’ (Oe6/Oe9) [17,18], ‘Picual’ (Oleur0.6.1) [13], and ‘Arbequina’ (Oe_Rao) [19], and a purported wild accession, *Olea europaea* subps. *sylvestris* (Oe451) [20]. Our full list of GBS samples and WGS samples is listed in Table 1. We removed 12 GBS samples from the initial 36-sample dataset that either failed to pass quality controls or for which no WGS data were available. The analysis was carried out on the remaining 24. The selection process is described in more detail in Section 4.5. These samples likely failed during the library preparation, although, there could be many reasons behind this problem. One of the most common is related to the quality of the DNA. Low DNA quality (e.g., due to impurities in the DNA extraction) reduces the efficiency of the restriction enzymes and leads to a partial digestion and reduction in the fragment population. Nevertheless, the low yield for the samples that failed (e.g., Barnea) indicates a bias in the amount of template used in the pool. Because the libraries were performed by an external service, it is difficult to assess where the problem was, but our guess is that it was related to the amplification step of some samples during the library preparation [35]. This would be an issue for a typical study of olive cultivar relatedness and would therefore need to be repeated. Fortunately, in this case the remaining samples were sufficient to compare GBS performance against WGS and the impact of assembly bias.

The assembly of GCA_002742605.1 (Oe451) used a whole-genome shotgun sequencing approach with the Illumina HiSeq 2000 platform to sequence *Olea europaea* var. *sylvestris* (wild olive). The assembly was performed using SOAPdenovo to generate a genome coverage of 220.0× [20]. For Oleur0.6.1, *Olea europaea* cultivar ‘Picual’ was selected as the genetic background [13]. The assembly process integrated Illumina HiSeq2500 and PacBio RSII sequencing to improve gap filling. *Olea europaea* cultivar ‘Farga’ based GCA_900603015.1/Oe6 [17] was assembled from Illumina HiSeq2500 with a genome coverage of 380×. Oe9 is an updated version of Oe6 [18], where a genetic map was used to anchor scaffolds to chromosomes. The most recently published is the *Olea europaea* cultivar ‘Arbequina’ (GWHAOPM00000000/Oe_Rao; NGDC) [19], which used Oxford Nanopore long-read sequencing and Hi-C data to construct chromosomes. The quality and completeness of each assembly was assessed before read mapping, SNP calling, and further analysis. Our comparison of genome assemblies was based on four metrics, (1) contiguity stats, (2) gene space completeness, (3) a k-mer completeness assessment using Merqury, and (4) the LTR Assembly Index (LTR) that evaluates the completeness of the genome using LTR retrotransposon elements (Table 2).

With respect to (1), assembly contiguity statistics were generated using a custom script (see Section 4.4). First, we note that total assembly size varied between genetic backgrounds; Oleur0.6.1, at 1.68 Gb long, was by far the largest assembly, and over 360 Mbs longer than both Oe6 and Oe9 (Table 2). Indeed, it was over 500 Mbps longer than Oe451 or the ‘Arbequina’ genome (Oe_Rao). Jiménez-Ruiz et al. proposed that the bigger size of the ‘Picual’-variety genome could be explained by the presence of a large number of duplicated DNA fragments coming from a very recent partial genome duplication event or artificially introduced repeat regions from assembly issues [18]. This comparison is interesting to observe because we are comparing the shotgun approach (Oe451, Oe6 and Oe9, Oleur0.6.1), where the genome is fragmented into reads of 250–800 bp [20], to sequencing with Pacbio first-generation RSII long reads (Oleur0.6.1), and the Oxford Nanopore third-generation long-read approach where reads are commonly 10–30 kb [41]. Shotgun approaches are very cost effective, but the process of sequence reassembly is more complicated than with long reads. The short reads make it difficult to correctly assemble repeat regions, particularly tandem repeats. This can also cause issues when using a genetic map to anchor to chromosomes, as the shorter read may not span a large enough section of the sequence to contain genetic markers. The trade-off is that on a per read basis, short reads are currently still more accurate and cheaper than long reads [41]. However, when only using short reads for an assembly, greater coverage is required, and this increases the number of errors in the dataset. Error correction and the filtering of low-quality reads is therefore an important step in the genome. We noted that Oe451, according to the Merqury assessment, had the highest QV score even with a 220× coverage, indicating careful management of the raw data during assembly. It is important to remember that each assembly will have its own unique set of errors introduced by sequencing or assembly issues, and thus any assembly should not be considered as a definitive sequence but rather, as stated in [42], only a working hypothesis.

Next, we checked (2) gene space completeness using Benchmarking Universal Single-Copy Orthologs (BUSCO) [43]. BUSCO genes are a set of ancestrally conserved genes used to estimate how complete a genome assembly is. Highly fragmented assemblies may contain greater percentages of missing or fragmented genes. The number of duplicated orthologs may be used as an indicator of possible errors that can occur when an assembly tool mistakenly assembles, or fails to assemble, reads into repeat regions. Nevertheless, polyploidy events and partial duplications may also lead to an increase in the number of duplicated genes detected by BUSCO, so often a deeper analysis needs to be carried out in order to distinguish a biological cause from a technical problem. In the publications for each of the olive tree genomes, BUSCO was used to evaluate assembly quality; however, the results were not comparable in terms of gene set database or BUSCO software version used. Thus, we ran BUSCO v.5 on all assemblies, using the eudicot_db10 gene set, which contains 2326 eudicot-specific genes. The results are different from those in the publications of the genomes used; however, this can be explained by the fact that the gene dataset’s specificity can vary greatly depending on which is used, and older versions of the gene datasets may be less complete. Using the eudicot_db10 dataset, we obtained an identical completeness score of 96.6% for Oleur0.6.1, Oe6, and Oe9, which indicates that a high proportion of the core gene space was captured by these assemblies [44].

In addition to a recent genome duplication event in ‘Picual’, there are signs of at least two older whole-genome duplications in Olive [20], so it is unsurprising to see high levels of duplicated genes here. However, the ‘Picual’ genome has 51.5% duplicated genes, more than twice any other assembly. This could be an indication of a technical error such as a failure in the consensus calling due to high levels of heterozygosity, or in this case, there is also evidence of a biological origin for a recent genome duplication. BUSCO gene sets cannot have taken into account very recently uncovered duplications, and so this high duplication percentage in Oleur0.6.1 is difficult to evaluate. The number of missing BUSCO genes was low in the ‘Picual’ and ‘Farga’ assemblies (1.4–1.5%), as was the percentage of fragmented genes (1.9–2%), indicating an overall high level of completeness. Oe451 scored the lowest using this dataset, with a completeness score of only 85.9, an indication that a large portion of the genome is still unassembled despite its pseudo chromosomes; this may be an issue for use in particular studies, such as synteny, or the discovery of candidate genes by QTLs, but may not be an issue for SNP calling as we had intended if the GBS sites are not in the unassembled regions. Oe_Rao scored 93.4 for completeness and had a lower percentage of duplicated genes compared to the ‘Farga’ and ‘Picual’ assemblies.

We assessed each of the assemblies using (3) Merqury, a k-mer-based method able to evaluate the quality and completeness of a de novo assembly without the need for a reference genome. Merqury uses a similar method to KAT in which high-quality sequencing reads are decomposed into k-mers datasets, then the k-mer sets are compared to the genome assembly. Merqury summarises the quality assessment with two values: a completeness score that measures the completeness of the assembly based in the k-mer populations of the assembled and the unassembled reads, and a phred-scaled consensus quality (QV) score that measures the error produced during the haploid sequence consensus calling of the assembly. Additionally, Merqury’s copy number spectra plots (Appendix A) allow for a visual inspection of unassembled reads and artificial duplications [45]. The evaluation of Oe_Rao could not be carried out correctly as Merqury requires short reads and the Oxford nanopore reads used in the assembly are not compatible with the process, and no Illumina reads were publicly available at the time of this analysis. The two ‘Farga’ assemblies and the ‘Picual’ assembly had an identical BUSCO gene completeness score, yet, using Merqury, the overall completeness of the ‘Farga’ and ‘Picual’ assemblies was vastly different. This indicates a high amount of reads which were never used in the final ‘Farga’ assemblies (Oe6/Oe9). Indeed, in the Appendix A plots Oe6 and Oe9, we can see that this is the case. This result may be partially explained by the co-sequencing of the fungus genome *Aureobasisium pullulans* with the olive samples, which led to an assembly of 18 Mb [17], but the K-mer multiplicity indicates that there was also an important proportion of the repetitive content unassembled. However, these three scored an equally high QV, indicating highly accurate consensus calling during contig assembly. Discounting the Oe_Rao results, Oe451 scored the lowest in terms of completeness and had the most k-mers found only in the assembly, but had the highest QV score. Sequencing error types and error rates vary with the sequencing technology [36,41,46]; these may cause contig misassembles and scaffolding, and thus affect the reliability of a genome assembly for use in the development of genotyping markers, breeding programs, or population studies. A good example of the impact of the genome completeness on olive genetic studies can be found in Kaya et al.: 51% and 75% of the SNPs with strong association signals were mapped to the Oe451 and Oe6 reference genomes, respectively. Although these percentages are not correlated with the estimation of the completeness from Table 2, they are a good indication that the quality of the genome can influence the usability in other analyses.

Finally, the assemblies were evaluated using the LTR Assembly Index (LAI). Genome assemblies based on short-read sequencing technologies such as Oe451, Oe6, and Oe9 presented lower values than the assemblies based on long-read sequencing technologies, Oleur0.6.1 and Oe_Rao, as was expected and previously described in the use of LAI for the assessment of genome quality [47]. It is interesting to note that the updated version of Oe6, Oe9, has a lower LAI (4.34 compared with 5.10), meaning that some of the transposable element genome space was lost or fragmented during the improvement. From all the genome assemblies used, only the Oleur0.6.1 had a LAI value over 10, such that it is considered a standard value for high-quality genomes with a good contiguity.

Because every sequencing project used a different technological approach, genetic background, and assembly methodology, there is necessarily a great deal of difference between them and their overall quality. While the limitations of short read length can impact the handling of repeat regions during scaffolding with a single short-read mapping at the incorrect or multiple regions, long-read technologies such as Oxford Nanopore and Pacbio have much higher error rates. Further, the choice of assembly tool, corrections, and the use of an alignment tool with and without available references can impact the quality of a genome assembly. Considering the information collected in Table 2, Oleur0.6.1 appears to be the most complete assembly with the highest QV score of the four we could test. However, it remains unclear if the high percentage of duplicated genes is an error or true genome duplication. Furthermore, this genome is not yet anchored to chromosomes, limiting its use in some studies.

### 2.2. Effect of Reference Genome Choice on Population Analysis

We explored the potential effect genome selection could have on a typical genetic diversity study such as a population analysis by mapping the same GBS reads to each of the available reference genomes, and performed the same population analysis on all the resulting SNP datasets. This analysis began with a quality control assessement of our dataset, which consisted of unprocessed reads from all 36 of our selected Olea europaea cultivars, amounting to over 326 million raw reads. After pre-processing, mapping, and SNP calling (see Section 4.5 and Appendix A) using three different genomes (Sylvestris/Oe451, Farga/Oe6, and Picual/Oleu r0.6.1), samples that had failed at one or more quality control checkpoints on all three genomes were removed from the rest of the analysis (see Section 4 and Section 4.5) (Appendix A). Removed samples are listed in Table 1. As mentioned previously, the GBS samples may have failed during library preparation, or sequencing and would normally need to be repeated; however, 24 samples are sufficient for the focus of this work. The remaining 24 samples were then additionally mapped to the two remaining genomes (Farga/Oe9 and Arbequina/Oe_Rao), before variant calling, SNP filtering, and population analysis.

#### 2.2.1. Analysis GBS Read Mapping and Variant Calling

Considering only the remaining 24 high-quality samples, the percentage of mapped reads did not vary much between assemblies; indeed, there was only a 2% difference separating the highest and lowest mapping genomes (Table 3). The average number of sites for each sample also showed very little variation, except for Oleur0.6.1, which had around 100,000 more sites, likely due to the increased number of duplicated genes in this assembly. It is important to highlight that a substantial difference was found in the number of variants called. Both Oe451 and Oe6 had double the number of variants called compared to Oe_Rao, and almost six times that of Oleur0.6.1. However, by far the most SNPs identified were from the improved ‘Farga’ genome, Oe9 with 23.7 million variants (Table 3). In terms of variants per loci, Oleur0.6.1 and Oe9 had, on average, almost triple the number of Oe451, Oe_Rao, and Oe6. The total number of SNPs called after filtering (see Section 4.5) was similar for all genomes. This massive drop off from Oe9 was most significantly explained by the 10,000 Kb thinning and 10,000 minimum quality score filtering steps. These two alone removed ~97% of total SNPs. Oleur0.6.1 was notable for having been left with half as many SNPs as the ‘Sylvestris’ and ‘Farga’ genomes. Perhaps the Oleur0.6.1 assembly contained several repeat regions which were incorrectly collapsed or, alternatively, reads may have been incorrectly assigned as a repeat(s), both of which could increase or decrease the number and frequency of SNPs, as multiple alleles from the same locus might be mistakenly identified as having come from different loci or vice versa. As can be seen in Table 2, there is indeed variation in the number of duplicated BUSCO genes and the k-mer duplications found across all of these genomes, indicating a difference in the assembly of repeat regions of each of the tested genomes. In such cases, it might be expected to also see variation in the levels of heterozygosity. However, heterozygosity per site was not significantly different, ranging from 0.31 to 0.36 (Table 3), making it difficult to interpret the source of this phenomenon. To test if collapsed regions were indeed the cause of this SNP count variation, we extracted the allele frequency of SNPs from three random samples (‘Grappolo’, ‘Manzanilla de Sevilla’, ‘Piñonera’) called from three of the genomes (Oe451, Oe6, Oleur0.6.1) and compared the frequency of alleles between 0.25 and 0.75 (Appendix A). Allele frequencies for an individual different from 0, 0.5, and 1, such as 0.25 and 0.75, may be an indicative of the collapse of four copies into one during the genome assembly. Once adjusted by percentage, we observed similar profiles, with most alleles tending towards 0.25 and a small peak around 0.5. However, from our sub-sampling we did not see any significant difference and so it remains unclear why using Oleur0.6.1 resulted in half the number of SNPs and Oe451.

#### 2.2.2. GBS Population Structure

The filtered SNPs obtained from each genome mapping were analysed using FASTSTRUCTURE (DARTR), ADMIXTURE (LEA), discriminate analysis of principle components (DAPC), and principal component analysis (PCA) to identify the distinct relationships among the cultivars (Appendix A).

In two previous studies, there were two distinct clusters identified among cultivars and a third cluster containing wild relatives [13,18]. In this work, the results from FASTSTRUCTURE identify two groups (K = 2, with cross entropy errors ranging 0.7 to 0.8) as being the most probable clustering for all GBS and WGS datasets (Appendix A). As wild olive samples were not included in the analysis, two groups are in agreement with the previously published data [13]. When comparing the results of the model-based clustering from ADMIXTURE and genetic-distance-based PCA clustering, it was noted to be in agreement also with FASTSTRUCTURE. The first three principal components of the PCA only explained ~30% of the variation. However, it was sufficient to see a clear separation of the different clusters, a result observed constantly throughout all our SNP datasets (Appendix A). In all cases, the first principal component (PC1) (~14%) separated the 24 cultivars into a group of 13 that could be described as primarily composed of eastern Mediterranean cultivars (‘Barri’, ‘Abou Kanani’, ‘Abou Satl Mohazam’, ‘Majhol-1013′, ‘Temprano’, ‘Verdial de Velez-Malaga-1′, ‘Uslu’, ‘Kalamon’, ‘Morrut’, ‘Mari’, ‘Picual’, ‘Manzanilla de Sevilla’, ‘Abbadi Abou Gabra-842′) and a group of 11 mostly northern Mediterranean cultivars (‘Mastoidis’, ‘Klon-14-1812′, ‘Grappolo’, ‘Mavreya’, ‘Piñonera’, ‘Leccino’, ‘Myrtolia’, ‘Menya’, ‘Manzanillera de Huercal Overa’, ‘Koroneiki’, and ‘Arbequina’), reflecting the complex history of olive domestication [11,48], and aligning with previously published results [13,18]. ADMIXTURE at K = 2 produced a similar grouping as the PCA (Appendix A). The levels of estimated admixture in each individual cultivar did not appear to vary depending on the assembly used. It is also interesting to note the near identical results regardless of the number of SNPs. Oleur0.6.1 had almost half the number of filtered SNPs as Oe451 or Oe9, but it appears that this still provides sufficient resolution to evaluate genetic relatedness using these analyses.

In general, the DAPC posterior membership was also estimated to be the same regardless of assembly used during mapping. We could see some variation in group membership when samples were compared by country of origin. There was largely agreement while comparing all genomes; however, we saw some discrepancies, such as in the Oe6, Oe9, and Oe_Rao genomes, which gave a near identical result in which the cultivar ‘Koroneiki’ of Greece showed a much higher probability of belonging to the Syrian genetic group. Indeed, this was more in agreement with the comparison of principal components 1 and 2, seen in all datasets belonging to the Syrian genetic group.

We calculated, for each dataset, genetic distances and constructed neighbor-joining distance trees using the R package POPPR (Appendix A). We can see in all cases there are two clear operational taxonomic units (OTUs), or clusters, defined in each tree (Figure 1) with a smaller third cluster of cultivars, which is more variable depending on the reference genome used. Samples in the third group were those which had higher levels of admixture (Appendix A) and were more difficult to assign. However, the two large groups were in agreement with the eastern Mediterranean and northern Mediterranean clusters identified in the PCA (Appendix A). The topology of each grouping varied, but only slightly, and only within an out. There was no placement of individuals in another cluster; even those with greater admixture were either in a seperate group or with cultivars of the same region. The topography of the trees produced from Oe451 and Oe6 were near identical, and even closer matched than Oe6 to its updated Oe9 version. The bootstrapping values were high in all cases with only 2–3 nodes falling below 50 on any tree. Syrian and Italian cultivars were the most consistent in terms of clustering when using any other assemblies. The cultivar ‘Mari’ was seen in all trees to cluster with Syrian samples, but, interestingly, the DAPC results indicate this same grouping only when using a ‘Farga’ or ‘Arbequina’ genome. This might be due to some genetic relationship between the genomes and the ‘Mari’ sample, such as a possible introgression with some of the SNPs.

Although there was great divergence in the number of SNPs associated with the use of each assembly, we saw little effect on the outcome of the population analysis. As observed in maize, reference genome selection can impact results [49,50]. Gage (2019) reported a reduction in the ability to robustly identify key loci of interest in a genome-wide association study (GWAS). Missing loci and inaccurate structural variations introduced during genome assembly may introduce reference bias, which may be more problematic in such cases. The impact of using any of the five selected assemblies in their current states appears to have been minimal in this population analysis, and indeed each reference genome provides confirmation of the results. However, it is still important to consider the vast differences in variants identified from each assembly and how they could impact other types of analysis. Using more than one reference assembly in this way can remove much of the bias introduced by many of the choices made during a genome sequencing project (see Section 2.1) and is something that could be repeated in future olive breeding projects to improve the reliability of identified structural variations.

### 2.3. Analysis of WGS Data

As the population analysis using GBS data showed that the selection of different reference genomes can make a difference in the phylogenetic relationship of one sample to another, but not in the overall grouping of samples, and no significant difference was seen in the levels of estimated admixture, we wanted to see if this was also the case with WGS data. Since WGS would likely generate many more variants and sites, any small differences may be diluted or potentially exaggerated. To process the raw whole-genome sequencing (WGS) read set, we followed the same steps as used for the raw GBS read set (see Section 4.5).

The processed reads were mapped to two assemblies, the purported wild type (Oe451) and the genome that scored the highest overall in our analysis (Oleur0.6.1). First, looking at the number of reads mapped to each, we saw that the WGS read set had a much higher number of mapped reads than the GBS set, as expected (Table 3). In both cases, the percentage of processed reads that was mapped was over 100%. Some double mapping was detected but not enough to cause issue. Variant calling using WGS data produced significantly more variants than GBS; 144 million from WGS-Oe451 and 128 million called from WGS-Oleur0.6.1 before filtering. It was not surprising to have so many more variants called using WGS reads because of the lower amount of missing data [36], but it was interesting to note that when we compared the total number of variants called from Oe451 and Oleur0.6.1 with the GBS data, there was a difference of 3.3 million between the two datasets before filtering. The difference between GBS-Oe451 and GBS-Oleur0.6.1 was almost double after filtering, but there was only a ~10% difference between WGS-Oe451 and WGS-Oleur0.6.1.

We next ran the same population analysis script as previously used for the GBS datasets. The FASTSTRUCTURE, PCA, admixture, and DAPC results were again very similar regardless of the genetic background of the reference genome. As before, there was some difference in the topography of trees but overall, the same OTUs were reproduced. Bootstrapping values were more consistently higher using Oleur0.6.1. The results are near identical to the GBS results, suggesting that it was possible to merge GBS and WGS datasets.

Here, we also compared the genomes that produced the highest and lowest number of SNPs. It is worth noting that, again, despite the higher number of SNPs coming from Oe451, there appeared to be no difference in the outcome of the analysis when using any of the reference genomes or SNP datasets. The reasons for this might be that SNPs provide limited information and so a certain threshold, in terms of reliability and number, is needed for an SNP dataset to be useful. The quality of the SNPs called and filtered from each reference genome was sufficient, even with the lower number of SNPs from Oleur.0.6.1, to extract the same level of detail. We also must consider that while five reference sequences were used, the genetic background of one was a purported wild type and all the others were Spanish cultivars. Our results show no difference when using either sequence, but this may change if a cultivar from another region with a more distinct genetic history is used.

### 2.4. Analysis of WGS/GBS Merged Data

The ability of an SNP dataset to differentiate between groups can be quite low because of their biallelic nature [51] and so it could be expected that increasing the number of SNPs from WGS should significantly improve the power of differentiation. However, our results show that GBS and WGS data performed almost identically, despite a difference of roughly 10 times the number of SNPs called using WGS data (Table 3). Given the similarity of the population analysis results, we wanted to know if it was possible to combine variants from GBS and WGS datasets. This would allow for more collaborative opportunities and the supplementation of smaller datasets with already available public data.

The same two genomes (Oe451 and Oleur0.6.1) were chosen for this experiment. The vcf files of GBS-Oe451 and WGS-Oe45 were merged together using bcftools and a list of shared sites (see Section 4), as were GBS-Oleur0.6.1 and WGS-Oleur0.6.1. The merged WGS/GBS-Oe451 file consisted of over 150 million variants and after filtering, 9537 biallelic SNPs, less than 1% of the total variants. The WGS/GBS-Oleur0.6.1 merged file contained only around 11 million variants and less than 1% of total variants were high-quality SNPs. This is in stark contrast to the difference seen with GBS-Oe451 and GBS-Oleur0.6.1. We selected only the SNPs from shared sites and did not allow for any missing data, so this would explain the massive drop off compared with either WGS or GBS after only filtering.

Again, we ran the same population analysis script as with all previous SNP sets. FASTSTRUCTURE found the mostly likely number of clusters, based on lowest cross-entropy, to be K = 18–19 for WGS/GBS-Oe451 and K = 20 for WGS/GBS-Oleur0.6.1 (Appendix A). This may be an overfit of samples due to there being two of each genotype (Appendix A) and the WGS samples being more like their GBS counterpart than anything else, creating the highest probability clusters. However, we could see that the change in cross entropy decreased between possible clusters past K = 2 and K = 3. The sample clustering by PCA was near identical for the WGS/GBS-Oleur0.6.1 and WGS/GBS-Oe451 datasets (Figure 2). Importantly, they were similar to the GBS and WGS only analysis with all the reference genomes. We saw that GBS and WGS samples closely grouped together, forming the same clusters seen previously. The only notable exception was GBS Arbequina. It seems likely that this sample was mislabelled at some point in the process, as it is the only sample that does not pair up with its WGS counterpart. DAPC analysis was analogous to all previous results, with the same groups being identified. Analysis by NJ trees (Figure 3) revealed no major differences in terms of OTUs, but we observed that Oleur0.6.1 had some very low bootstrapping values and Oe451 had the better bootstrapping values, more closely resembling the WGS-only tree.

## 3. Conclusions

The results of this work indicate that GBS and WGS sequencing data are highly comparable for population structure analysis. Certainly, more information can be obtained from WGS, and great advances have been made to reduce its cost, but GBS remains much more affordable. For small sampling sizes WGS maybe be preferable, but a large-scale genotyping project can quickly become too expensive for most labs; we show that GBS is a cost-effective alternative capable of providing near identical results and identifying the same genetic relationships at a much lower cost. Further, the ability to successfully combine the two sequencing methods opens opportunities to mine data from a wider range of sources. This would also be a significant cost-saving approach to consider for expanding existing olive tree WGS datasets and past genotyping projects, and allow for collaborations between research groups that have used either WGS or GBS genotyping methods. Although the sequencing cost is decreasing, it is not feasible to re-sequence hundreds of accessions to study the structure of the population or to identify some unknown accessions.

With respect to our comparative analysis between the different reference genomes used, the Oleur0.6.1 sequence may be the most accurately assembled, but lacking chromosomes can limit its application. However, we could not properly test the ‘Arbequina’ assembly using Mercury’s best practices and therefore accurately compare it to the others in all aspects of our quality assessment. Any differences between the current existing genome assemblies are so small that we can say that for SNP-based genetic profiling at least, all are certainly suitable. Thus, the availability of genetic maps and chromosomes could potentially be a more important consideration when choosing which to use in future projects. Furthermore, we show that while the genetic background of the reference genome plays a role in the possible number of variants identified, there was little overall effect on population structure, genetic clustering, or analysis of genetic relationships. As more reference genomes of existing cultivars are rendered to the chromosome level, it might well be necessary to compare performance before choosing which to use. The current assemblies are of a purported wild type and three Spanish cultivars, so it is still possible that the genetic background may yet be an important factor if using cultivars from other regions as the reference.

Our results highlight the advantages of GBS while at the same time bringing to the table the possible limitations. GBS is a commonly used technique for other crops, but it has not been routinely implemented in olive breeding. Traditional breeding needs these types of tools to accelerate the development of new varieties able to face the important challenges olive production is currently facing.

## 4. Materials and Methods

### 4.1. Plant Material and Genomic DNA Extraction

In accordance with Belaj et al. (2012), and maximising the genetic diversity in a reduced number of genotypes, 36 olive tree cultivars were selected from the World Olive Germplasm Collection of the Andalusian Institute of Agricultural and Fisheries Research and Training (WOGBC) located in Córdoba Spain (Table 1). Total genomic DNA was extracted from fresh leaves using the Illustra DNA extraction kit Phytopure GE Healthcare (UK) in accordance with the protocol described in the manufacturer’s instructions. To ensure high-quality DNA was used for sequencing, the purity was measured with a Qubit 2.0 Fluorometer Life Technologies NY (USA). DNA concentration was then normalised to 20 ng/µL. The DNA had a minimum 260/280 ratio of 1.8.

### 4.2. GBS Library Construction and Sequencing

The DNA were sent to DISMED, and the libraries were prepared by BGI. DNA samples were digested with ApeKI (New England Biolabs, Ipswitch MA) for 2 h at 75 °C, and T4 ligase (New England Biolabs, Ipswitch MA) was used to ligate to sticky ends, one of 36 unique “barcodes”, and the “common adapter”. Samples were incubated at 22 °C for 1 h and heated at 65 °C for 30 min. A set of 36 digested DNA samples, each with a different barcode adapter, were obtained. A total of 7 µL of each component of this set was combined in a unique sample and purified in a final volume of 50 µL with a commercial kit (QIAquick PCR Purification Kit; Qiagen group (Germany), according to the manufacturer’s instructions. The result of this library was amplified in 50 µL, containing 5 µL of pool DNA fragments, 1× Taq Master Mix New England Biolabs (UK), and 12.5 pmol each of PCR primers, the sequences of which were: PCR primer: 5′-AATGATACGGCGACCACCGAGATCTACACTCTTTCCCTACACGACGCTCTTCCGATCT and PCR primer: 5’-CAAGCAGAAGACGGCATACGAGATCGGTCTCGGCATTCCTGCTGAACCGCTCTTCCGATCT, containing complementary sequences for amplifying the fragments of DNA with ligated adapters. The PCR conditions were a primer step of 5 min at 72 °C; 98 °C for 30 s; 25 cycles of 98 °C for 30 s, 65 °C for 30 s, and 72 °C for 30 s; and a final extension step at 72 °C for 5 min. The library was purified as above (in a final elution of 30 µL) and 1 µL was used for the quality evaluation of fragment sizes. The library was considered suitable for sequencing if adapter dimers were minimal (~128 pb in length) and the majority of the other DNA fragments were between 170 and 350 bp. Paired-end sequencing of one 48-plex library per channel was performed on an Illumina HiSeq2000 Analyzer by BGI Genomes.

The sequences of the barcode adapter were: 5-ACACTCTTTCCCTACACGACGCTCTTCCGATCTxxxx and 5-CWGyyyyAGATCGGAAGAGCGTCGTGTAGGGAAAGAGTGT, where ‘‘xxxx’’ and ‘‘yyyy’’ are the barcode and barcode complement, respectively. The second, or ‘‘common’’ adapter sequence was shared among all samples and consisted of an ApeKI-compatible sticky end: 5´-CWGAGATCGGAAGAGCGGTTCAGCAGGAATGCCGAG and 5´-CTCGGCATTCCTGCTGAACCGCTCTTCCGATCT.

### 4.3. WGS Library Construction and Sequencing

Raw sequencing data were obtained directly from the authors of Jiménez-Ruiz et al. (2020). Re-sequencing of all varieties was performed by 2 × 150 paired-end sequencing with Illumina HiSeq 4000, and all sequencing was carried out at the Duke Center for Genomics and Computational Biology (Durham, NC, USA). Raw data are available at NCBI BioProject ID: PRJNA556567.

### 4.4. Sequence Assembly Assessment

The five available assemblies used were *Olea europaea* subsp. *sylvestris* (version GCA_002742605.1/Oe451; NCBI), *Olea europaea* cultivar Picual (version Oleur0.6.1; https://genomaolivar.dipujaen.es/db/ (accessed on 2 February 2020)), the *Olea europaea* cultivar Farga (version GCA_900603015.1/Oe6; NCBI) [17] and its updated version (GCA_902713445.1/Oe9; NCBI) [18], and *Olea europaea* cultivar Arberquina (GWHAOPM00000000/Oe_Rao; NGDC) [19]. The same three forms of quality assessment were carried out on all the assemblies: contiguity, gene space completeness, and a k-mer-based evaluation.

Assembly and contiguity were calculated using a custom script; FastaSeqStats (https://github.com/aubombarely/GenoToolBox/blob/master/SeqTools/FastaSeqStats (accessed on 10 June 2013). N50 is a commonly used marker of sequence contiguity, counting the number of bases in the shortest fragments needed to span 50% of the genome. N90 is the same information at 90% of the total genome. Average sequence length provides similar information as the N50. L50 is the smallest number of contigs needed to cover 50% of the genome; L90 provides the same information at 90%.

Benchmarking Universal Single-Copy Orthologs (BUSCO) genes were used to evaluate the completeness of the assembly by searching for a list of known ancestrally conserved genes within the reference genome. All assemblies were assessed using BUSCO v.5 with the eudicot_db10 gene set containing 2326 eudicot-specific genes.

Merqury v1.3, (https://github.com/marbl/merqury (accessed on 10 January 2021) a k-mer-based method, was used to evaluate both quality and completeness. Using a similar method to KAT, high-quality sequencing reads were decomposed into k-mers datasets, then the k-mer sets were compared to the genome assembly. Merqury generates a k-mer dataset from the Illumina short-read sequencing data used in the genome sequence assembly. By decomposing the original sequencing reads into k-mers, Merqury can count how many times each k-mer appears in the assembly, as well as k-mers from the original Illumina reads not incorporated at all or that appear only in the assembly. These k-mer data were used to generate a completeness score and a phred-scaled consensus quality (QV) score, along with copy number spectra plots to visually inspect the assembly for unassembled reads and artificial duplications [45]. The Illumina short-read datasets were either downloaded from the same location as the assembly or kindly sent by the genome assembly curator. In the case of the ‘Arbequina’ assembly, only long-read sequencing was performed, which was unsuitable for use with Merqury.

LTR Assembly Index (LAI) was estimated using the LTR_Retriever tool v2.9.0 with the default parameters [47].

### 4.5. Read Processing, Mapping, Filtering, and Variant Calling

Raw reads were first processed and then mapped only to Oe451 to assess quality, using the pipeline described below and in Appendix A. Samples with low SNP count can affect the overall number of SNPs available for analysis due to the percentage of missing data encountered during the VCF filtering (see Section 4 and Appendix A). Before removing poorly performing samples, the full process was repeated with two other assemblies (Oleur0.6.1 and Oe6) to confirm the results (Appendix A). Total SNPs called from Oe451, Oe6, and Oleur0.6.1 after filtering were initially in the range 300–900. To increase this number further, low-quality samples were removed. The choice of samples to be included or excluded was based on performance at different stages of processing and the amount of missing data. To achieve this, the number of reads in each sample was counted and there were three clear groups, those with only 50,000 or fewer reads, a second group with 600,000 or fewer, and the majority of samples were within the range of 2–20 million reads. Samples with 50,000 or fewer reads were removed from further analysis, as this is too low to yield any useful information, along with any samples having less than 50% of the average number of sites per sample.

This reduced the sample set from 36 to the current 24, but significantly increased the number of total SNP loci shared across all samples. Sequence read mapping was then performed on all five genome assemblies to evaluate the influence that a particular assembly has on SNP discovery and on a population analysis using the following steps.

After sequencing, the GBS raw reads were demultiplexed by BGI Genomics. Next, the raw reads were processed with FASTQ_MCF v1.05 [52] to remove Illumina adapters and reads with a phred-scaled quality score of less than 30 and/or shorter than 50 bases. After trimming, the paired reads were aligned to each of the five genome assemblies; Oe451, Oe6, Oe9, Oleur0.6.1, and Oe_Rao, using BWA (Burrows–Wheeler Alignment Tool) v0.7.17-r1188t with default parameters. Prior to mapping, each reference genome was indexed using BWA. The output file of the mapping was in an unsorted SAM format, and these were converted to bam and sorted to save space and increase SNP calling efficiency using SAMTOOLS v1.7 [53]. Once sorted, the bam files were merged into a single bam file with BAMADDRG (https://github.com/ekg/bamaddrg (accessed on 14 April 2018).

Variants were called with FREEBAYES v1.3.1-16-g85d7bfc [54] using a custom script; MultiThreadFreeBayes, (https://github.com/aubombarely/GenoToolBox/tree/master/SNPTools/MultiThreadFreeBayes (accessed on 15 May 2018) this script allows FREEBAYES to run faster by using multiple threads on several scaffolds at the same time. Finally, the variant file (VCF) was filtered with VCFTOOLS v0.1.15 [55] using the following parameters: retain only biallelic SNPs, remove indels, a minimum read depth of 5 with a minimum mean depth of 20, a minimum SNP quality of 1000, no missing data, and an MAF of 0.05. Finally, SNPs were thinned to 1 per 10,000 Kb.

To merge the GBS and WGS datasets, we used the BCFTOOLS v1.7 [53] merge function. The WGS VCF file contained many more sites than the GBS, so a bed file containing all sites from the GBS dataset was supplied using the argument regions-file. This was carried out to reduce the final file size, as any site with missing data would eventually be removed during the filtering steps. After the files were merged, they were filtered using the same parameters as above.

### 4.6. Population Analysis

Each SNP dataset was analysed in the same way, with the same script to estimate population structure and genetic diversity from each of the datasets. The R script Olea_pop.R with commands and notations is available at https://github.com/frieljames/Olive_WGS_GBS (accessed on 1 April 2021). Appendix A shows an overview of the R programming pipeline. As a summary of the script, we cross analysed population structure in our datasets with the use of R and three different methods: (1) STRUCTURE [56], a Bayesian-based clustering method assuming Hardy–Weinberg equilibrium and linkage equilibrium between loci within populations, (2) discriminant analysis of principal components (DAPC) [57], which uses sequential K-means and model selection to infer genetic clusters, and (3) principal components analysis (PCA), a method that uses genetic distance to infer clusters. To begin, we used FASTSTRUCTURE [58], part of the DARTR package, a faster and more resource-efficient method of running STRUCTURE. The LEA package [59] performed a STRUCTURE analysis on the SNP data to infer the mostly likely genetic clusters based on allele frequency and clustering probability. ADMIXUTRE was estimated by running LEA’s snmf functional analysis for the estimation of ancestral populations (K) of 1–20. This estimates the admixture coefficient in the selected K range using sparse Non-Negative Matrix Factorisation algorithms. The output of this was used to estimate the admixture of groups at the largest K value. To perform a DAPC and a PCA, we used the R package ADGENETv.1.3-1 [60]. DAPC is different to STRUCTURE in that it does not rely on model assumptions or prior information; instead, it uses a multivariate method with sequential K-means and model selection to infer genetic clusters and assign individuals to clusters. To ensure that maximum variance was being used while attempting to avoid an overfit of the data, DAPC was performed using the suggested optimal number of principle components (PCs) for each dataset (Appendix A). These were identified as the number of PCs with the highest a-score, predicted by the optim.a.score() function, which selects an evenly distributed number of PCs in a pre-defined range, computes an a-score for each, and then interpolates the results using splines. For the PCA, the package ADGENET took a genlight object of SNP data to generate a genetic distance matrix for use in the PCA clustering.

Further estimation of possible ancestral populations was achieved with a neighbour-joining (NJ) approach, again based on allele frequencies. This was carried out with the R package POPPR [61]. The POPPR function aboot() allowed for the construction of a dendrogram with 100 bootstrapping support and obtained genetic distance using bitwise.dist, a method that calculates the fraction of different sites between samples equivalent to Provesti’s distance. The tree was visualised in R with the package APE [62] using the function plot.phylo(). Afterwards, the tree was exported in the newick format to generate figures using FigTree v1.4.4 [63].

Finally, genetic diversity was calculated using the gl.basic.stats () function with the R package DARTR [64]. Selection and neutrality were estimated by calculating the number of segregating sites, nucleotide diversity, Watterson’s theta, and Tajima’s D. These values were generated using POPGENOME [65]. Additionally, population diversity was estimated using expected and observed heterozygosity, Fst and Fis values, for each of the assigned groups (POPGENOME) (Appendix A).

## Figures and Tables

**Figure 1 plants-10-02514-f001:**
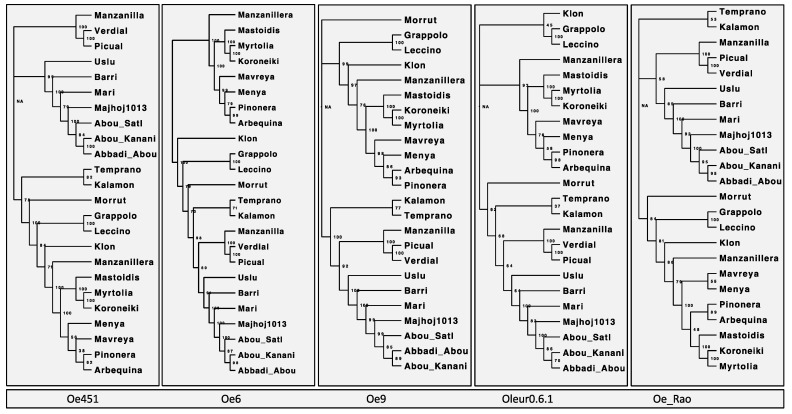
Neighbour-joining trees constructed using GBS data. Comparison of the neighbour-joining (NJ) trees constructed from SNPs called using GBS data mapped to each of the reference genomes. Using the R package Poppr, the genetic distance between each sample using Provesti’s distance and NJ trees is plotted with a bootstrapping support of 100. The genome used to generate each is labelled above the corresponding tree.

**Figure 2 plants-10-02514-f002:**
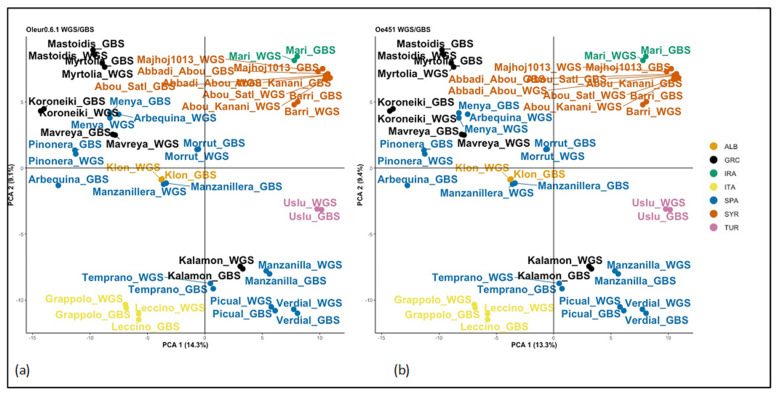
PCA clustering of WGS/GBS datasets. This shows the clustering of WGS and GBS merged samples. Principle components 1 and 2 are shown for each. (**a**) PCA of merged data with reference genome Oe451; (**b**) PCA of merged data with reference genome Oleur0.6.1.

**Figure 3 plants-10-02514-f003:**
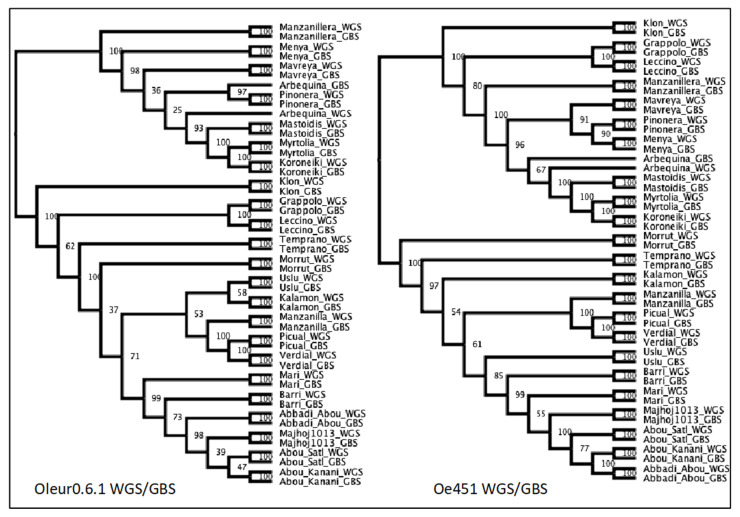
Neighbour-joining trees constructed from merged WGS and GBS data mapped to the Oleur0.6.1 and Oe451 reference genomes. Genetic distance between each sample using Provesti’s distance and NJ trees plotted with a bootstrapping support of 100.

**Table 1 plants-10-02514-t001:** List of *Olea europaea* cultivars sequenced using GBS and available WGS data.

Sequenced Cultivar	Country of Origin	Country Code	Available Data
Klon-14-1812	Albania	ALB	GBS/WGS
Kalamon	Greece	GRC	GBS/WGS
Koroneiki	Greece	GRC	GBS/WGS
Mastoidis	Greece	GRC	GBS/WGS
Mavreya	Greece	GRC	GBS/WGS
Myrtolia	Greece	GRC	GBS/WGS
Mari	Iran	IRA	GBS/WGS
Grappolo	Italy	ITA	GBS/WGS
Leccino	Italy	ITA	GBS/WGS
Arberquina	Spain	SPA	GBS/WGS
Manzanilla de Sevilla	Spain	SPA	GBS/WGS
Manzanillera de Huercal Overa	Spain	SPA	GBS/WGS
Menya	Spain	SPA	GBS/WGS
Morrut	Spain	SPA	GBS/WGS
Picual	Spain	SPA	GBS/WGS
Piñonera	Spain	SPA	GBS/WGS
Temprano	Spain	SPA	GBS/WGS
Verdial de Velez-Malaga-1	Spain	SPA	GBS/WGS
Abbadi Abou Gabra-842	Syria	SYR	GBS/WGS
Abou Kanani	Syria	SYR	GBS/WGS
Abou Satl Mohazam	Syria	SYR	GBS/WGS
Barri	Syria	SYR	GBS/WGS
Majhoj-1013	Syria	SYR	GBS/WGS
Uslu	Turkey	TUR	GBS/WGS
Removed Cultivars			
Chemlal de Kabylie	Algeria	DZA	GBS
Megaritiki	Greece	GRC	GBS
Shengeh	Iran	IRA	GBS
Barnea	Israel	ISR	GBS
Frantoio	Italy	ITA	GBS
Forastera de Tortosa	Spain	SPA	GBS
Llumeta	Spain	SPA	GBS/WGS
Picudo	Spain	SPA	GBS/WGS
Jabali	Syria	SYR	GBS
Maarri	Syria	SYR	GBS
Majhoj-152	Syria	SYR	GBS
Dokkar	Tunisia	TUN	GBS/WGS

**Table 2 plants-10-02514-t002:** Genome assembly and contiguity statistics.

*Cultivar*	Wild Olive	Farga	Farga	Picual	Arbequina
Assembly Stats	Oe451 GCA_002742605.1 (Unver et al., 2017)	Oe6 GCA_900603015.1 (Cruz et al., 2016)	Oe9 GCA_902713445.1 (Julca et al., 2020)	Oleur0.6.1 (Jiménez-Ruiz et al., 2020)	Oe_Rao GWHAOPM00000962 (Rao et al., 2021)
Chromosome assembly	Yes	No	yes	no	yes
Assembly size (Gb)	1.14	1.32	1.31	1.68	1.1
Scaffolds	41,256	11,038	9753	9174	962
Longest seq (Mb)	46.03	2.58	36.44	4.14	68.07
Shortest seq (bp)	452	500	500	1017	10,772
Average seq length (Kb)	27.69	119.47	135.00	183.12	1146.54
N90, number of seq	3410	3099	2019	4503	320
L90 (Kb)	23	111	116	86,918	279,924
N50, number of seq	23	901	162	1145	11
L50 (Mb)	12.57	0.44	0.73	0.41	42.60
% BUSCO complete	85.9	96.6	96.6	96.6	93.4
% BUSCO duplicated	13.4	23.6	23.6	51.5	18.3
% BUSCO fragmented	5.9	1.9	2.0	1.9	2.1
% BUSCO missing	8.2	1.5	1.4	1.5	4.5
Merqury completeness	72.62	78.39	78.73	89.41	NA
Merqury QV	43.00	34.78	35.68	33.04	NA
Merqury error	5.00867 × 10^−5^	0.00033	0.00027	0.00050	NA
LTR_Retriever LAI	4.14	5.10	4.34	11.52	8.80

Assembly stats generated using custom script. Gene space completeness assessed using BUSCO v5 with eudicot_db10 dataset. Completeness, repeat regions, and sequencing data incorporation evaluated with Merqury v1.3 and Illumina short-read-derived k-mers. LTR Assembly Index (LAI) was estimated using the LTR_Retriever tool v2.9.0 with the default parameters.

**Table 3 plants-10-02514-t003:** GBS and WGS read mapping and variant calling.

Cultivar	*O europaea* var. *sylvestris*	Farga	Farga	Picual	Arbequina
	Oe451 GCA_002742605.1 (Unver et al., 2017)	Oe6 GCA_900603015.1 (Cruz et al., 2016)	Oe9 GCA_902713445.1 (Julca et al., 2020)	Oleur0.6.1 (Jiménez-Ruiz et al., 2020)	Oe_Rao GWHAOPM00000962 (Rao et al., 2021)
Total GBS reads mapped (M)	294.29	298.86	299.56	300.28	297.58
Total WGS reads mapped (M)	8767	NA	NA	8789	NA
% of total GBS reads	94.4	95.8	96.0	96.3	95.4
% of total WGS reads	100.4	NA	NA	107.14	NA
Avg number of sites GBS	325,680	350,581	348,652	432,644	340,715
Avg number of sites WGS	493,618	NA	NA	1,073,623	NA
GBS SNP before filtering	7,722,425	7,415,201	23,651,384	4,401,892	1,346,358
GBS SNP after filtering	13,343	12,507	13,410	7023	10,177
GBS unfiltered SNPs per site	1.01	0.90	2.91	2.39	0.89
GBS filtered SNPs per site	0.02	0.02	0.02	0.01	0.02
Average Het/site GBS	0.36	0.34	0.35	0.31	0.35
Average Het/site WGS	0.42	NA	NA	NA	0.34
WGS SNP before filtering	144,579,296	NA	NA	128,172,089	NA
WGS SNP after filtering	119,190	NA	NA	114,169	NA
WGS/GBS SNP before filtering	150,952,260	NA	NA	11,679,534	NA
WGS/GBS SNP after filtering	9537	NA	NA	4004	NA

## Data Availability

The novel GBS data produced for this study can be found under the NCBI Bioproject accession PRJNA750928. WGS data were already published, associated with the Bioproject accession PRJNA556567.

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
