# Peer review of "Comparative Analysis of Genotyping by Sequencing and Whole-Genome Sequencing Methods in Diversity Studies of Olea europaea L."

_plants, 2021, doi:10.3390/plants10112514_

Round 1
Reviewer 1 Report
The manuscript “Comparative Analysis of Genotype-by-sequencing and Whole Genome sequencing Methods in Olea europaea L.” compares whole genome sequencing and genotype by sequencing method to estimate genetic relatedness in Olive genotypes. This is an interesting work and their findings would provide a clue to follow a suitable sequencing and genotyping method as well as reference genome and genetic maps for olive genetics and genomic studies. The manuscript is very well written and some of the minor oversights are mentioned for corrections.
- The abstract is not clear. Please add the aim, objective and significance of the MS
- Please speculate about the reasons for the obtained results. The discussion needs to improve accordingly.
- Figure 2: The text mentioned near to PCA dots aren’t readable. Please change the color which is readable to every section of audience.
- Line 121: Olea europaea should be italicized. Check the genes and scientific names in the text for italicizing according to convention.
- Line 170: What is the significance of ‘parentages’ here?
- Correct the typos and grammatical mistakes (For eg. Line 36: Itis should be It is, line 160: greatercoverage, Line 293: FASTTRUCTURE should be FASTSTRUCTURE).
- The English of the whole article has to be checked carefully to eliminate linguistic errors.
Author Response
Dear reviewer
Thank you very much for your suggestions which will help us to improve the manuscript significantly. We have attached a file with the reply and modifications of all your suggestions

Reviewer 2 Report
Thank you for the opportunity to review the manuscript titled, "Comparative Analysis of Genotype-by-sequencing and Whole Genome sequencing Methods in Olea europaea L."
The research is really interesting and provides a piece of good information for the readers. However, I have some minor comments listed here.
- I think picture/pipeline/bullet format would be a good way to represent the sequencing/analysis technique.
- Please shed light on the importance and advantages of conducting this research with clear future directions.
- Spacing error: Line 20, 24, 26, 36, 56, 143, 162, 295, 312, 461, 466, 527, 540, 548
- There are some grammatical errors and the font size needs to be consistent.
- Please introduce all acronyms.
Author Response

(The authors gave the same response as above.)

Reviewer 3 Report
The field of olive research possess considerable genetic and genomic resources that require close inspection to determine how to get the best benefit from the knowledge accumulated. Friel and colleagues proposed an innovative way of mining curated GBS and WGS datasets (genotyping by sequencing and whole genome sequencing) for variant analysis against the available olive genome assembly drafts. The authors carefully evaluated all the genome assemblies and emphasized the importance of selecting the best genome assembly and the impact of it in variant calling. Overall, this is an important contribution to the olive research field and an original idea in the genetics analysis area.
The manuscript is well written, although seems like the conversion to *.pdf format merged hundreds of words (highlighted in the file attached).
A minor comment regarding the wet bench methodology is noticed: It would be helpful to complete some specific details around the quality of DNA used for GBS library preparation and note if these were outsourced (DISMES?) or home-made.
The only thing I would like to see discussed is the possibility of generating GBS and WGS libraries simultaneously (prepared and evaluated separately, before pooling), and consider pooling ratios ([GBS] > [WGS], or equimolar?), its effect on increasing library complexity that would ease some issues seeing when sequencing reduced representation libraries, and also the benefits of having exactly the same individuals/DNA preps used for generating these data sets.

Author Response

(The authors gave the same response as above.)

Reviewer 4 Report
The authors compared the performance of GBS or WGS methods in Olea europaea L, using population structure analysis as an indicator. This article demonstrates very little difference from either method. However, these results are unsurprising, with limit insightful information provided. Several questions or comments are as below:
- Table 1, there are 3 materials both have GBS and WGS data. As author mentioned, they may fail to pass quality controls, but the quality parameters should be given.
- Line 137, “Hi-Cdata” should be “Hi-C data”, lacking a spacing. There are many cases lack that in the current manuscript, please check carefully accordingly.
- Line 287-317, a plot of cross entropy errors for the ancestor number “K” is needed for exact K selection. Both FASTSTRUCTURE and ADMIXTURE are needed.
- Figure 1 and Line 369-385, the aim of this paper is to compare the performance of GBS or WGS methods. To this end, the showing of contrastive NJ-trees constructed by GBS or WGS SNPs is necessary. While, the authors only showed the NJ-tree for GBS results. Additionally, There are lots of another analysis to compare the performance of GBS or WGS methods, such as FASTSTRUCTURE, PCA, admixture and DAPC, and authors take up much space to descript them, however, there are no any tables or figures. This is not appropriate.
- Line 386, “2.3. Analysis of …” should be “2.5. Analysis of …”.
- Normally, the WGS SNPs should, to some extend, overlapping with GBS SNPs, what’s the accordance?
- The conclusions of this article were exaggerated. As the results showed, the authors only could conclude that the performance of GBS or WGS was comparable for population structure analysis. For other cases of use, such as GWAS, high-density linkage mapping and so on, maybe not applicable.
Author Response

(The authors gave the same response as above.)

Reviewer 5 Report
This research considers the use of cost effective GBS markers instead of costly WGS analyzes in diversity studies. The results detailed in this comparison are valuable for both population research and breeding purposes encompassing large populations or hundreds of samples where WGS would be out of reach for a small to medium budgets. The authors also compares the quality of the available Olea europea reference genomes and demonstrates their effect on the outcome of the diversity analysis as the reference genome of choice for read mapping.
Major points:
- Reference bias: With the increasing availability of de novo genomes, the bias introduced in both forward and reverse genetics is better understood nowadays and can greatly affect trait mapping and gene identification. See
-
Gage JL, Vaillancourt B, Hamilton JP, Manrique‐Carpintero NC, Gustafson TJ, Barry K, Lipzen A, Tracy WF, Mikel MA, Kaeppler SM, Buell CR, Leon N (2019) Multiple Maize Reference Genomes Impact the Identification of Variants by Genome‐Wide Association Study in a Diverse Inbred Panel. Plant Genome 12:180069
-
Della Coletta R, Qiu Y, Ou S, Hufford MB, Hirsch CN (2021) How the pan-genome is changing crop genomics and improvement. Genome Biol 22:1–19
-
- From the results presented by the author, the reference bias does not seem to affect estimation of genetic relationships which is an interesting and important conclusion, however it also means that the results presented here are relevant only for these studies - while this is repeatedly addressed in the paper I would also consider adding it to the title, something along the lines of Comparative Analysis of Genotype-by-sequencing and Whole Genome sequencing Methods in diversity studies of Olea europaea L
- For clarity, I think a supplementary table should be made available with stats per each of the 24 cultivars detailing % reads mapped and the filtered number of SNPs.
- Throughout the paper there are a lot of grammar mistakes between plural and singular, and some spelling mistakes.
- The materials and methods section is well written and very detailed.
- I suggest moving the conclusion paragraph before the materials and methods.
Specific comments:
- Table 2: Add LAI indexing for assembly quality
Ou S, Chen J, Jiang N (2018) Assessing genome assembly quality using the LTR Assembly Index (LAI). Nucleic Acids Res 46:e126
- Table 3: inconsistency of formatting – Thousands separator comma, decimal point where > 999 is irrelevant. It is also unclear which stat is average per cultivar (of the remaining 24?) – e.g. 294.23 reads total reads mapped across all samples?
- Line 110: The genetic background is indeed and important consideration - see reference bias.
-
Line 209: Anticipated misspelled
- Lines 206 – 212: I agree with the authors assessment. I would suggest to remove this sentence. List NA in Table 2 for Oe_Rao and explain here that Merqury best practices details the use illumina reads that were unavailable for this sample.
-
Line 255: The author discusses little variation between samples, but the stats are missing – maybe add supp table.
-
Line 268: LAI index stat may help clarify repeat completeness in assemblies.
-
Line 368: Delete either 'would significantly' or 'should improve'.
-
Line 549: Github link leads to empty repository.
Author Response

(The authors gave the same response as above.)

Round 2
Reviewer 4 Report
I do not have additional question now!